# Beyond Vision: An Overview of Regenerative Medicine and Its Current Applications in Ophthalmological Care

**DOI:** 10.3390/cells13020179

**Published:** 2024-01-17

**Authors:** Francisco J. Santa Cruz-Pavlovich, Andres J. Bolaños-Chang, Ximena I. Del Rio-Murillo, Guillermo A. Aranda-Preciado, Esmeralda M. Razura-Ruiz, Arturo Santos, Jose Navarro-Partida

**Affiliations:** 1Tecnologico de Monterrey, Escuela de Medicina y Ciencias de la Salud, Monterrey 64849, Mexico; fspavlovich@gmail.com (F.J.S.C.-P.); andres.jbc98@gmail.com (A.J.B.-C.); ximena.drm@gmail.com (X.I.D.R.-M.); esmi44@outlook.com (E.M.R.-R.); arturo.santos@tec.mx (A.S.); 2Instituto de Oftalmologia Guillermo Avalos Urzua, Guadalajara 44260, Mexico; tony-aranda@hotmail.com

**Keywords:** regenerative medicine, stem cells, exosomes, scaffolds, gene therapy, ophthalmology, clinical trials

## Abstract

Regenerative medicine (RM) has emerged as a promising and revolutionary solution to address a range of unmet needs in healthcare, including ophthalmology. Moreover, RM takes advantage of the body’s innate ability to repair and replace pathologically affected tissues. On the other hand, despite its immense promise, RM faces challenges such as ethical concerns, host-related immune responses, and the need for additional scientific validation, among others. The primary aim of this review is to present a high-level overview of current strategies in the domain of RM (cell therapy, exosomes, scaffolds, in vivo reprogramming, organoids, and interspecies chimerism), centering around the field of ophthalmology. A search conducted on clinicaltrials.gov unveiled a total of at least 209 interventional trials related to RM within the ophthalmological field. Among these trials, there were numerous early-phase studies, including phase I, I/II, II, II/III, and III trials. Many of these studies demonstrate potential in addressing previously challenging and degenerative eye conditions, spanning from posterior segment pathologies like Age-related Macular Degeneration and Retinitis Pigmentosa to anterior structure diseases such as Dry Eye Disease and Limbal Stem Cell Deficiency. Notably, these therapeutic approaches offer tailored solutions specific to the underlying causes of each pathology, thus allowing for the hopeful possibility of bringing forth a treatment for ocular diseases that previously seemed incurable and significantly enhancing patients’ quality of life. As advancements in research and technology continue to unfold, future objectives should focus on ensuring the safety and prolonged viability of transplanted cells, devising efficient delivery techniques, etc.

## 1. Introduction

Regenerative medicine (RM) is defined as the development of biological substitutes that restore, maintain, or improve the function of tissues or whole organs [1]. It focuses on the use of stem cells, tissue engineering, scaffolds, soluble molecules, gene therapy, and the reprogramming of cell and tissue types [1,2]. The term RM was first mentioned by Leland Kaiser in 1992 but was best defined by William Haseltine in 1999, who described it as an emerging field that merged different subjects such as tissue engineering (TE), cell transplantation, stem cell biology, biomechanics, prosthetics, nanotechnology, and biochemistry [3].

Nowadays, RM has been successfully used in orthopedics for articular cartilage regeneration, in dermatology for improving nasolabial fold wrinkles and in skin regeneration for severe burns, among other applications [4]. Moreover, in ophthalmology, the interest in RM has been increasing thanks to its applicability in the field and the physiological advantages that the eye possesses. The fact that the eye is easily accessible and immunologically privileged make it an ideal site to evaluate RM therapies [5]. Multiple studies have evaluated the efficacy and security of RM in treating both anterior and posterior segment diseases, in both pre-clinical and clinical studies, with mixed results [5,6,7].

The objective of this review is to present a general panorama of the current strategies in the field of RM: cell therapy, exosomes, scaffolds, in vivo reprogramming, organoids, and interspecies chimerism. Additionally, it aims to envelop the applications of RM in the field of ophthalmology.

## 2. Regenerative Medicine Strategies

### 2.1. Cell Therapy

Cellular therapy is based on the utilization and transplantation of autologous or allogenic cells into a patient with the objective of treating or managing a medical condition. It encompasses both the practices of stem cell and genetically engineered non-stem cell-based unicellular or multicellular therapies [8]. 

A stem cell is a functionally non-committed, primitive cell with a myriad of functions and possible specialization pathways that is able to differentiate into more physiologically specialized cells and repair injured tissue. Once they are inoculated into a given milieu, they proliferate and begin to specialize down a given cellular lineage [9]. The most important characteristics of stem cells are self-renewal, clonality, and potency. Self-renewal refers to the capacity to proliferate extensively; clonality to the ability to arise from a single cell; potency, also called differentiation potential, refers to the capability to differentiate into different cell types [10]. 

Regarding potency, five stem cell categories exist: totipotent, pluripotent, multipotent, oligopotent, and unipotent cells, with the first having the highest differentiation potential and then orderly decreasing [10]. Pluripotent and multipotent stem cells have been the most extensively studied for clinical use due to their potential to differentiate into multiple cell lineages and promote tissue repair.

On the other hand, depending on their origin, stem cells can be classified as embryonic stem cells (ESCs), fetal stem cells (FSCs), and adult stem cells (ASCs). These cells can be obtained by multiple isolation methods (e.g., density gradient centrifugation, conditioned expansion media, flow cytometry) from the donor tissues and are later generally expanded (with some exceptions) for injection directly into the damaged organ/tissue or also into the systemic circulation [11].

Even though cell therapies show promise as novel methods for treating disease, there are issues regarding the use of stem cell therapies which continue to be studied. Firstly, immunorejection is a considerable problem to note, as well as genetic instability, which could lead to carcinogenesis in the host. It has been reported in mice that teratomas and other related tumors could develop secondarily to the use of pluripotent stem cell transplantation [10,12,13,14]. Another issue that has limited the success of cellular therapies is the poor localization, retention, and survival of the transplanted cells at the injury site [15]. Finally, ethical concerns regarding the use of stem cells have also been noted, especially with ESCs.

Following this, a broad view of the different cell therapies is presented, classifying the different types of stem cells based on their origin. A brief representation of the most important cell therapies and their administration in ophthalmology can be seen in Figure 1.

#### 2.1.1. Embryonic Stem Cells

ESCs are pluripotent stem cells that are derived from the inner cell mass of the blastocyst and can be identified by several surface markers, including CD324, CD90, CD117, CD326, CD9, and CD29, among others [16,17]. ESCs also express a wide variety of nuclear transcription factors, including factors essential to maintain pluripotency, including Oct-3/4, Sox2, KLF4, and NANOG [17].

ESCs cultured in the right conditions have an indefinite proliferative life span and can maintain pluripotency [18]. Moreover, ESCs have been used for a wide array of applications, from the study of monogenic inherited diseases and chromosomal aberrations to clinical trials trying to repair damaged tissues and organs [19]. For instance, ESCs have been used to treat spinal cord injury, Parkinson’s disease, diabetes, and retinal disease, among others. ESCs provide several advantages, including scalability, being xeno-product free, and expeditious administration, which is especially useful for the treatment of incurable and terminal diseases [20].

Despite these advantages, the use of ESCs for research and clinical purposes has been surrounded by ethical and religious controversies. Previously, the extraction of ESCs involved the destruction of the embryo; nowadays, this limitation has been overcome. Nonetheless, the development of other stem cell therapies has diminished the interest in ESCs [19,20].

#### 2.1.2. Fetal Stem Cells

Fetal tissue- and extra-fetal tissue-derived stem cells can all yield fetal stem cells (FSCs). FSCs are an intermediate between ESCs and adult stem cells, exhibiting several properties shared by these cell groups. The earlier in gestation they are derived, the more primitive features they show [21]. FSCs are less controversial in terms of ethics and safety when comparing them with ESCs since fetuses are never created with the sole purpose of stem cell harvest [21]. FSCs are generally obtained from cadaveric, 6 to 12 week fetuses originating from miscarriages, stillbirths, ectopic pregnancies, and elective abortions [22]. FSCs also exhibit several advantages in contrast to adult stem cells; first, they are less likely to be rejected by transplant recipients, second, they show greater plasticity, and lastly, they are easier to cultivate and show faster proliferation rates [21,23].

##### Fetal Tissue-Derived Stem Cells

Fetal hematopoietic, mesenchymal, and neural stem cells are the most promising for RM applications [24,25]. Hematopoietic stem cells (HSCs) can be readily obtained from fetal blood, bone marrow, and liver, all of which proliferate at higher rates than adult HSCs [25,26]. Moreover, fetal mesenchymal stem cells (fetal MSCs) have been isolated from BM, pancreas, kidney, lung, liver, and other tissues. They have greater developmental potential than their adult counterparts, as well as better immunosuppressive and immunomodulatory capacities [25,27,28]. Regarding fetal neural stem cells, they have been found to be distributed mainly in the fetal hippocampus, subventricular zone, striatum, and cortex and can give rise to neurons, astrocytes, and oligodendrocytes, showing promise in the treatment of neurodegenerative diseases [22,29]. Interestingly, tissue-specific fetal stem cells can also be found. A relevant example are retinal progenitor stem cells, which give origin to all the different types of retinal cells [30].

##### Extra-Fetal Tissue-Derived Stem Cells

Extra-fetal stem cell sources include the placenta and its different layers, amniotic fluid, and the umbilical cord. These tissues are not involved in as many ethical squabbles as the previous types of stem cells as they are normally disposed of after parturition. The use of these cells also eliminates the surgical risks involved with stem cell isolation from adult tissues, making them excellent stem cell reservoirs [31].

Placental-derived stem cells (PDSCs) share some characteristics with ESCs and can be obtained from the distinct layers of the placenta, namely, the amnion, chorion, and decidua [31,32,33,34]. From the amnion, two types of amniotic stem cells can be found: epithelial cells (from the epiblast) and mesenchymal cells (from the hypoblast). From the chorion, chorionic MSCs (from the inner mesoderm) and chorionic trophoblast cells (from the outer layer of trophoblast) can be found. Finally, aptly named decidual MSCs derive from the decidua [35].

Amniotic fluid (AF) has been used for decades as a tool for prenatal diagnosis of genetic diseases and recently has been found to also be a reservoir of stem cells. AF is the only extraembryonic tissue that can be harvested before parturition, making it extremely useful for the development of in utero cell-based therapies [31,36]. There are two subsets of stem cells in the AF: AF-stem cells (AFSCs) and AF-mesenchymal stem cells (AFMSCs) [37]. The former express CD117, while the latter express mesenchymal cell surface markers, such as CD90 and CD105 [31,37].

Finally, the umbilical cord has been widely used as a reservoir of stem cells, mainly of HSCs and MSCs. In the umbilical cord blood, more HSCs per volume unit can be found than in adult BM, making it an attractive alternative. Moreover, umbilical cord HSC transplant carries a lower risk of mismatches and, consequently, a lower graft-versus-host disease (GVHD) risk [38,39]. On the other hand, surrounding the vessels is a connective tissue called Wharton’s jelly, which is rich in proteoglycans. The extracellular matrix of this connective tissue is abundant in umbilical cord MSCs (UCMSCs), which can differentiate into a variety of mesodermal cell types, including fat, bone, and skeletal muscle [31]. These cells exhibit stem cell markers like c-Kit, NANOG, OCT4, and SOX2 in their expression. Interestingly, UCMSCs have special capabilities for chondrogenic differentiation [40].

#### 2.1.3. Adult Stem Cells

Also called somatic stem cells, adult stem cells are undifferentiated and can be found among differentiated cells in the whole body after development. These cells enable healing, growth, and the replacement of cell losses that occur daily and have a restricted range of differentiation options [16]. One of the most important advantages of adult stem cells are that autologous cells do not raise issues of rejection or ethical controversies [10,41].

##### Hematopoietic Stem Cells

HSCs are immature, multipotent cells that can be found in the peripheral blood and BM. These cells can develop into all types of blood cells, including lymphoid lineage and myeloid-lineage cells [42]. HSCs can be identified by the expression of several markers; amongst the most important, CD34, CD133, CD90, CD49f, and CD201 can be found [43,44]. Furthermore, homologous and allogenic HSC transplant have been used for decades to replace lost or dysfunctional bone marrow HSCs for the treatment of several conditions, such as chemotherapy- or radiotherapy-induced myeloablation and diverse hematologic and genetic disorders [42,45]. Interestingly, although debated, it has been shown that HSCs can contribute to the regeneration of non-hematopoietic cells in different disease models [46].

##### Mesenchymal Stem Cells

MSCs are multipotent, non-hematopoietic stem cells with extensive proliferative ability and with the potential to differentiate into the various mesenchymal lineages [47]. Because of their lack of a specific surface marker, the International Society for Cellular Therapy established the minimum criteria to be met to classify a cell as an MSC, as follows: adherence to plastic under standard culture conditions; expression of CD105, CD73, and CD90; lack of expression of CD45, CD34, CD14 or CD11b, CD79a, or CD19 and human leukocyte antigen-DR; and differentiation into osteoblasts, adipocytes, and chondroblasts in vitro [48].

It has been proposed that MSCs arise from pericytes; nonetheless, other studies have suggested that pericytes, even though very similar to MSCs, are a distinct cell population or even a subpopulation of MSCs in a perivascular location [49]. MSCs can be isolated from almost every tissue that is vascularized including bone marrow (BM), adipose tissue, umbilical cord, dental pulp, and skin. These cells additionally function as paracrine and secretory centers at injury sites in the body [2]. Among the tissue sources, BM and adipose tissue are the most studied and prevalent in MSC clinical trials.

Regarding bone marrow-derived MSCs (BM-MSCs), they were the first MSCs to be described. In humans, BM-MSCs may be isolated from BM aspirated from the sternum, vertebral body, iliac crest, and femoral shaft [50]. The most important characteristics that BM-MSCs can offer are their significant potential for protecting ischemic tissues at risk, modulating inflammation, autoimmunity, and specially promoting tissue regeneration [2,51]. BM-MSCs more readily differentiate into the osteogenic and chondrogenic lineages and their differentiation capacity is reduced by age [52,53]. No standardized isolation and culture expansion method exists for MSCs, but typically consist of the use of density centrifugation to separate the mononuclear cell fraction from the other bone marrow constituents [49]. This mononuclear cell fraction contains T cells, B cells, monocytes, HSCs, endothelial progenitor cells, and MSCs and, in fact, this cell mixture of mononuclear cells has been used as the transplantation product in different trials [49,54,55]. Finally, to isolate MSCs, this mononuclear cell fraction is plated in culture flasks, taking advantage of the plastic adherence property of MSCs [49].

One of the disadvantages of the use of BM-MSCs is the low stem cell cellular yield from the aspirate; in 1 mL of aspirate, approximately 6 × 10^6^ cells can be isolated, from which only 0.001% to 0.01% are MSCs [56]. Therefore, utilizing BM-MSCs for therapeutic purposes typically requires large amounts of BM. Nonetheless, different expansion and characterization methods have been developed in order to overcome this issue, thus allowing for a greater number of BM-MSCs to be obtained [57,58]. Another issue is that isolation and expansion of BM-MSCs in culture require close monitoring to avoid cell senescence, but methods addressing this have also been established [59]. In addition, although considered a safe procedure, in the literature it can be found that the aspiration of BM commonly from the sternum and posterior iliac crest has been documented to be fatal, resulting in cardiac tamponade, hemorrhage, and osteomyelitis [60,61].

Respecting adipose tissue-derived stem cells (ASCs), adipose tissue is a very dynamic and complex endocrine organ composed mainly of extracellular matrix, mature adipocytes, and stromal vascular fraction (SVF), which includes adipose tissue-derived stem cells (ASCs), preadipocytes, pericytes, endothelial cells, smooth muscle cells, fibroblasts, hematopoietic cells, and immune cells [62,63]. Depending on the isolation method, harvest site, and age of the donor, the viable nucleated SVF cells are estimated to be 2–6 × 10^6^ cells/g of adipose tissue. From those, approximately 3.5 × 10^5^ to 1 × 10^6^ are ASCs, a much higher yield than the BM-MSCs obtained from 1 g of bone marrow [62,64,65]. The most common approach for ASC isolation is via the enzymatic digestion of adipose tissue using collagenase II, followed by centrifugation and isolation of the SVF. The cells from the SVF are then cultured, permitting the isolation of the ASCs as they are the only cells from the SVF that show plastic adherence [65,66].

Due to their mesodermal origin, ASCs can differentiate into adipogenic, chondrogenic, and osteogenic cells, with a special competence for adipogenic differentiation [53]. Regarding ASC markers, these cells have shown to be a heterogeneous cell population, so no unique marker has been identified, but in comparison with BM-MSCs, ASCs show a high expression of CD49d and low expression of Stro-1 [53]. As any other MSC, ASCs express the typical mesenchymal markers previously mentioned and are negative for hematopoietic antigens [65]. ASCs have a unique secretome that makes them an excellent mediator of tissue regeneration, containing several cytokines, growth factors, morphogens, chemokines, and extracellular vesicles [67]. Moreover, ASCs have been shown to be more resistant to oxidative stress-induced senescence, hypoxia-induced apoptosis, have a more potent proangiogenic activity, and higher telomerase activity when compared with BM-MSCs [68].

The clinical application of ASCs has demonstrated success in the treatment of multiple diseases and injury states, including cardiovascular, inflammatory bowel disease, diabetes mellitus, kidney and spinal cord, bone and craniofacial reconstruction, liver cirrhosis, multiple sclerosis, systemic lupus erythematosus (SLE), and GVHD [62].

##### Induced Pluripotent Stem Cells

During the period of 2006–2009, three independent research groups reported successful genetic reprogramming of somatic cells to stem-like cells and coined the term induced pluripotent stem cells (iPSCs) [2]. The Nobel laureate Yamanaka and his group were the first to successfully reprogram mouse embryonic fibroblast cells in 2006 [69]. A year later, human skin fibroblast-derived iPSCs were reported [70].

iPSCs are generated from adult cells by overexpression of embryonic genes or the core transcription factors named Yamanaka factors, including OCT4/3, SOX2, Klf4, and c-Myc [2,69]. The overexpression of these factors is obtained by integrative (meaning that the genes integrate into the hosts’ DNA) viral and non-viral vectors or by repeated exposure to non-integrative viral and non-viral vectors until the pluripotent state is reached [71,72]. Because of their ability to self-renew, proliferate, and produce germ line competent chimeras (meaning that they can integrate into a blastocyst), iPSCs are almost identical to ESCs [2]. iPSCs have the additional advantages of having easy accessibility and expandability, and they can be induced to differentiate into hundreds of cell types [73]. Moreover, iPSCs are derived from adult cells, not embryos, overcoming the major ethical restrictions involving ESCs. Additionally, the iPSC technology has integrated into innovative technologies such as gene editing and three-dimensional organoids, which have greatly boosted efforts in disease modeling, drug discovery, and cell therapy [74].

Unfortunately, the use of iPSCs has given rise to several safety concerns: the possibility for dangerous clones to emerge, the risk of contamination from still undifferentiated cells, genomic instability, potential to form teratomas, and the possibility of epigenetic aberrations. All are issues that need to be primarily solved before translating this therapeutic tool into a bedside treatment [75,76].

##### Induced Tissue-Specific Stem Cells

Nowadays, there are several unresolved issues related to the use of iPSCs for clinical therapies; with this in mind, researchers have recently successfully generated tissue-specific stem cells and coined the term induced tissue-specific stem cells (iTSCs) [2,77].

iTSCs are an intermediate cell between adult differentiated cells and iPSCs. Reprogramming of these cells has been achieved by transient, non-integrative viral vector overexpression of the core transcription factors and were firstly generated from mouse pancreas and liver somatic cells. These cells were shown to generate insulin-producing cells and hepatocytes, respectively, while maintaining self-renewal potential [72]. It was demonstrated that after the reprogramming, an epigenetic memory is inherited from the parental cells, explaining the predisposition to differentiate into their cells of origin [72]. Moreover, iTSCs have not shown any tumorigenic potential, overcoming the risks involved in pluripotent stem cell transplantation [78].

##### Very Small Embryonic-Like Stem Cells

More than 15 years ago, a new kind of pluripotent stem cell residing in adult tissues was discovered. These cells seemed like the solution to the long search for a pluripotent stem cell free of ethical concerns (such as those involved with ESCs) and technical issues (like in iPSCs generation) [79,80]. Very Small Embryonic-Like Stem Cells (VSELs) are small, 5–7 μm, early development stem cells similar in morphology with the cells from blastocysts’ inner cell mass, hence their name. Additionally, they express several ESC markers such as SSEA, Oct-4, Nanog, and Rex1 [80]. VSELs have been isolated from several tissues, including BM, gonads, and the umbilical cord [80,81]. Under physiological conditions, VSELs are in a quiescent state, which is explained in part due to their protection from insulin/insulin-like growth factor stimulation as a result of the epigenetic modification of paternally imprinted genes; in different circumstances, cancers would spontaneously occur [82,83,84,85]. Expansion of VSELs in vitro was an important issue when these cells were first isolated, as VSELs did not divide and expand in culture; nonetheless, nowadays it is possible to expand them upon exposure to epigenetic regulators like nicotinamide and valproic acid [86]. VSELs share with the other types of pluripotent stem cells the ability of differentiating into the three germ layers; additionally, VSELs have several unique properties including the capacity to differentiate into gametes in vitro and differentiate into adult cell types in vivo. Moreover, VSELs do not form teratomas and do not integrate into blastocysts as ESCs and iPSCs do [84].

One of the biggest disadvantages of VSELs is their rarity and difficulty for isolation; in fact, this issue has brought a debate regarding if these cells even exist or if they really are stem cells [87,88,89,90]. Several independent groups have been able to isolate VSELs, but a lot of doubt and criticism still exists [80,91]. The skeptical argument is that more standards, characterization, and common assays are needed and also that the commercial interest in these cells represents an issue itself; they find worrisome that some clinical trials using VSELs are on the way when there is not yet a clear consensus on their possible benefits or even their existence [88,91].

### 2.2. Extracellular Vesicle Therapies: Exosomes

Exosomes are endosome-derived lipid bilayer spherical vesicles that are secreted by all eukaryotic cells. These vesicles are functional vehicles that carry proteins, lipids, and nucleic acids from the parenting cell and act as a paracrine method of intercellular communication for processes such as immune responses, signal transduction and antigen presentation [92]. As seen in Figure 2, exosomes measure from 40 to 150 nm in diameter and contain thousands of proteins, micro RNAs (miRNAs), messenger RNAs (mRNA), non-coding RNAs (ncRNA), transfer RNAs (tRNA), ribosomal RNAs (rRNA) and rarely DNA, and also several lipids [93,94]. Because of their endosomal origin, exosomes include membrane-associated proteins, including tetraspanins (e.g., CD9, CD63, CD81, and CD82, which serve as exosome surface markers), MHC I and MHC II, several heat-shock proteins (e.g., Hspa8, Hsp60, Hsp70, Hsp90), GTPases (EEF1A1, EEF2), and proteins important for the vesicles’ biogenesis (Alix and TSG101) [93,94,95]. Exosomes also contain metabolic enzymes (e.g., GAPHD), cytoskeletal proteins (e.g., actin), and carrier proteins such as albumin [93]. Exosomes are derived from invaginating buds in the plasmatic membrane, which endocyte several extracellular and plasmatic membrane proteins, lipids, and metabolites. These endocytic vesicles then fuse, forming early endosomes (EEs) which soon after mature into late endosomes (LEs). LEs next form multiple membrane invaginations that selectively encapsulate proteins and nucleic acids, generating intraluminal vesicles (ILVs). LEs are subsequently termed multivesicular bodies (MVB), containing numerous ILVs which are then secreted as exosomes after the fusion of MVBs with the plasmatic membrane [96,97]. Following their secretion, exosomes interact with the recipient cells through their surface receptor molecules and ligands. This induces endocytosis or fusion of the vesicle, allowing its contents to be internalized and produce different effects on the recipient cell. For instance, exosome-derived miRNAs have been shown to regulate gene expression by inhibiting specific target genes [95,98].

Exosomes can be readily obtained by a diverse variety of techniques (e.g., ultracentrifugation, ultrafiltration) and have been extensively studied with regard to their diagnostic and therapeutic potential [99]; furthermore, of late, exosomes have been identified as a potential player in regenerative medicine as it has been identified that stem cells, specially MSCs, play a major role in regeneration not only by cell replacement but also by secretion of exosomes that promote tissue repair and regeneration [95]. With this discovery in mind, the idea emerged that a free-cell therapy, containing MSC-derived exosomes, could be an effective, safe alternative to cell-based therapies to promote regeneration, avoiding the risks associated with direct stem cell transplantation [95]. Multiple in vitro and in vivo studies have been performed to analyze the regenerative potential of MSC-derived exosomes, especially for the delivery of specific miRNAs and proteins for different organs, with favorable results. Regarding therapies for nerve regeneration, it has been shown that its use can boost neurite outgrowth, induce neuronal differentiation, and enhance angiogenesis and neurogenesis, while attenuating neuroinflammation and boost neuronal survival and proliferation, among other promising effects [100,101,102,103]. It was also demonstrated that they could promote the survival of retinal ganglion cells and the regeneration of their axons, making exosomes an attractive therapy in ophthalmology [104]. Other in vitro and in vivo studies testing exosomes for their regenerative potential have been performed in other organs, such as the heart, skin, bone, muscle, cartilage, liver, and kidneys, also with optimistic results [95].

Multiple clinical trials involving the use of exosome-based therapies for specific conditions are currently on the way, with a few completed showing varying results; even so, the clinical translation of these therapies for regenerative medicine has yet to be further explored [94,105].

### 2.3. In Situ Regeneration: Scaffolds

Tissue engineering (TE) is based on the use of cells, scaffolds, and growth factors to replace or regenerate damaged tissues/organs. It can be used in RM along with the other RM therapies to induce in vivo regeneration [106]. One of the main strategies in TE is the application of scaffold-based methods to promote in situ regeneration, as can be seen in Figure 3 [107]. In this approach, biomaterials are used to create support structures for cells which will then form the desired tissue. The scaffold is a porous, fibrous, or permeable three-dimensional (3D) construct that provides a template for the regeneration of the affected tissue, while concurrently promoting cell attachment, proliferation, and ECM generation [108]. These constructs should be biomimetic, biodegradable, non-cytotoxic, immunologically inert, and provide mechanical stability [2,108,109]. Scaffolds are classified based on their composition into metals, ceramics, natural/synthetic polymers, and composites, all with specific applications. For instance, ceramic, metal, and composite scaffolds have been mainly studied for hard tissue regeneration, such as bone and teeth, while polymers are used for both soft and hard tissue engineering [106,110,111,112]. Specifically, polymers can be further classified as natural or synthetic. Natural polymers can be proteins (i.e., collagen, gelatin, fibrinogen, silk), polysaccharides (i.e., glycosaminoglycan, hyaluronic acid, chitosan), or polynucleotides (DNA, RNA), while synthetic polymers can be made of degradable (i.e., polyesters, polylactones) or non-degradable components (i.e., polyether) [108,112,113,114,115]. They can be engineered by multiple methods, with freeze-drying, electrospinning, and 3D bioprinting the most common [111,116,117,118]. Depending on the desired purpose, scaffolds can be of different forms, with, hydrogels, sponges, films, and fibers the most used [15,119]. Moreover, scaffolds can be both acellular and cellular. For instance, acellular scaffolds carry bioactive materials (i.e., growth factors) and when implanted integrate into tissues and generate microenvironments that promote repair [106,120,121]. On the other hand, cellular scaffolds are cell-coated and provide the cells with an appropriate microenvironment for growth, overcoming some of the challenges involved in stem cell transplant alone; in fact, transplanted stem cells show poor viability, which is in part explained by the poor microenvironment in the affected tissues, which does not allow cellular retention and engraftment [122,123].

### 2.4. In vivo Reprogramming

Reprogramming can provide specific cell types for regenerative medicine applications to replace tissues lost or damaged by degenerative disease and injury [124]. In vivo reprogramming includes the use of gene therapies, epigenetic reprogramming, and gene editing. Most reprogramming procedures are performed in vitro, but the long-awaited use case for reprogramming for regenerative medicine applications increases the need for in vivo reprogramming [125].

#### 2.4.1. Gene Therapy

Gene therapy is the ability to improve genetics by correcting mutated genes or site-specific modifications for the purpose of therapeutic treatment. It is defined as a therapeutic strategy that transfers DNA into a patient’s cells to correct a defective gene or gene product to treat diseases that cannot be treated with conventional drugs [125]. A schematic view of the basis of gene therapy can be seen in Figure 4. This therapy has been made possible by advances in genetics and bioengineering that have made it possible to manipulate vectors to deliver extrachromosomal material to target cells [126]. The US Food and Drug Administration (FDA) defines gene therapy as products “that mediate their effects by transcription and/or translation of transferred genetic material and/or by integrating into the host genome and that are administered as nucleic acids, viruses, or genetically engineered microorganisms”. It also states that “the products may be used to modify cells in-vivo or transferred to cells ex-vivo prior to administration to the recipient” [127].

Methods for classifying gene therapy include sorting by the class of disease (genetic disease versus complex acquired disorder), by the characteristics of the gene delivery vehicle (integrating versus nonintegrating), and by whether the vector is administered in vivo (directly into the patient) or ex vivo (in cultured cells taken from the patient that are subsequently transplanted back) [128].

In ex vivo gene therapy, the patients’ cells are collected, cultured, modified, and transplanted back to the patient. Compared to traditional allogenic transplantation, this cell-based gene therapy does not need a histocompatible donor and avoids GVHD. Retroviruses, such as lentiviruses, are usually used as vectors to deliver a normal copy of a specific defective gene into the genome of transplanted cells [129]. Ex vivo gene delivery has been successfully applied in the treatment of adenosine deaminase-associated severe combined immunodeficiencies (Strimvelis^®^), β-thalassemia (Zynteglo^®^), and large B-cell lymphoma (Yescarta^®^ and Kymriah^®^) [130,131,132].

Comparatively, in vivo gene delivery avoids the practical hurdles of cell collection, culture, modification, and transplantation seen in ex vivo cell-based gene therapy. It directly delivers a normal copy of a specific defective gene into the target cells through local or systemic delivery. Adeno-associated viruses (AAV) are the main vectors used for in vivo gene therapy. AAV vectors are single-stranded DNA (ssDNA) non-integrating vectors, meaning that the delivered DNA is not integrated into the genome of target cells as retroviruses do. Non-integration reduces the risks of insertional mutagenesis but limits long-term expression from AAV vectors in target cells [129]. AAV-mediated in vivo gene delivery has been successfully applied in the treatment of familial lipoprotein lipase deficiency (Glybera^®^), retinal dystrophy (Luxturna^®^), and spinal muscular atrophy (Zolgensma^®^) [132,133,134,135,136].

#### 2.4.2. Epigenetic Reprogramming

Epigenetic mechanisms have the property of making changes in gene expression patterns without producing modifications in the DNA sequence. Specific epigenetic expression programs take an important role during mammalian development as the zygote experiences several epigenetic events to be capable of differentiating into all cell types of the body. These include DNA methylation, histone modifications, and the use of noncoding RNAs (e.g., miRNAs, long non-coding RNAs) [137,138]. Epigenetic reprogramming has been mostly used in in vitro scenarios to reprogram cells into other cell types. Nonetheless, a handful of studies have approached real, in vivo epigenetic reprogramming. These studies have taken a special interest in the use of reprogramming for the reversal of the effects of aging as it has been found that epigenetic modifications, specifically DNA methylation of CpG islands in several loci, are strongly correlated with this phenomenon [139,140,141].

One of the most important methods for cell reprogramming is by the ectopic expression of transcription factors using viral vectors (e.g., adenovirus, lentivirus). Transcription factors can induce differentiation of one adult cell to another through epigenetic modifications in vitro. For example, Ascl1, Brn2, and Myt1 can reprogram fibroblasts into neurons via chromatin remodeling [142]. Moreover, it has been observed that the Yamanaka factors (essential to create iPSCs) induce several epigenetic modifications to reprogram adult cells into pluripotent cells. Nonetheless, these factors are not only useful for pluripotent stem cell reprogramming; for instance, the in vivo introduction of these transcription factors, while controlling their expression in a cyclical manner, has been demonstrated to reset the DNA methylation age in old mice, thereby promoting cells into a “younger” state while avoiding the erasure of cellular identity (as in iPSCs) and extending the mice lifespan [140,141,143]. Moreover, this younger state reprogramming caused the recovery of vision in mice glaucoma models and also natural ageing mice visual loss [143].

miRNAs have also been shown to exhibit the capacity to reprogram cells alone or in combination with other transcription factors. Attributable to their somewhat small size, miRNAs can be conveniently delivered in cells for initiating reprogramming [137]. As a matter of fact, epigenetic reprogramming can be also used to create iPSCs; for example, miRNAs such as miR-302 is known to facilitate the reprogramming of human skin cells to iPSCs [144].

Additionally, small chemical molecules have also been used for reprogramming. Of special interest, DNA methyltransferase, histone deacetylase, histone methyltransferase, and histone demethylase inhibitors can be found. For instance, these inhibitors have been found to increase the efficiency of the Yamanaka factors for iPSCs reprogramming [137,145,146].

#### 2.4.3. Gene Editing

Gene editing refers to the technology in which various DNA repair methods are adopted by the introduction of programmable nucleases into target tissues or cells to achieve functional repair or specific functions [147,148]. Gene editing is different from gene therapy as it is intended to manipulate the existing DNA sequence, in contrast to gene therapy whose aim is the addition of new genes [149]. In therapeutic genome editing, a nuclease-encoding gene is delivered into target cells via viral vectors. Alternatively, a nuclease could be delivered into target cells in the form of mRNA or protein with the help of nanoparticles or lipids [147]. The delivered programmable nucleases include meganucleases, zinc finger nucleases (ZFN), transcription activator-like effector nucleases (TALEN), and clustered regularly interspaced short palindromic repeat (CRISPR)-associated nuclease Cas9 (CRISPR-Cas9) [147]. These nucleases introduce DNA double-strand breaks (DSBs) at specific genomic loci where the defective target gene localizes. The delivered nuclease can achieve its therapeutic effect by the correction/inactivation of prejudicial mutations, addition of protective mutations, the introduction of therapeutic transgenes, or viral DNA severance [147]. After the DSBs, endogenous repair machinery for either non-homologous end-joining (NHEJ) or homology-directed repair (HDR) are mobilized to the DSB location, completing the gene editing procedure [129,147,150].

### 2.5. Organoids

Organoids are 3D organ-like structures in vitro that retain the specific framework and properties of the original organs in the body [151]. Organoids can be initiated from pluripotent, fetal, and adult stem cells and when allowed to differentiate in culture, these cells demonstrate the ability to self-assemble into structures that reflect important aspects of the tissues for which they are intended [152]. In fact, with the proper 3D scaffold and biochemical factors, self-organizing tissue-specific organoids have been generated, such as intestine, liver, kidney, heart, retina, and brain [153,154]. Compared with usual cell cultures, organoids highly reproduce the functional, biological, and structural complexity of an organ, recreating the tissues’ local architecture, morphology, and numerous organic interactions taking place in vivo [155,156].

Organoid development gives researchers the ability to grow isogenic tissues from patient biopsies for use in transplants [157]. Overall, preclinical data support positive engraftment of organoids after transplantation as these 3D structures have been observed to integrate, mature, vascularize, and develop specific target tissue physiological functions. Moreover, it has been shown that these organoids produce differentiated and functional cells capable of interacting with other host cells after transplantation [156].

Organoid technology, although a promising candidate for regenerative therapies, is still in the early stages of development and several challenges still need to be addressed, such as the relative immaturity of the organoid grafts, incomplete integration, and heterotypic cell interactions [156].

### 2.6. Interspecies Chimerism

Human–animal chimeras offer the potential to produce human tissues and organs in other species for transplantation purposes. This technology has attracted a lot of interest in the context of the potential prospect of using interspecies chimeras in basic and translational research [158]. It basically consists of the introduction of human autologous pluripotent stem cells, mainly iPSCs, to an animal blastocyst (called “blastocyst complementation”), where the iPSCs would then contribute to the chimera formation with the resulting organs being patient-specific and immunized for transplantation. The basis of interspecies chimerism is the precept that mammalian development is noticeably conserved throughout species [159]. After reprogramming for naive pluripotency, human pluripotent stem cells show a very low ability to generate interspecies chimeras. Whether this is because they a priori lack attributes of chimeric competence or because naive iPSCs cannot colonize embryos of distant species remains to be seen [160]. Promisingly, mouse–rat live chimeras have already been successfully created, with successful chimerized organs containing both mouse and rat cells [161].

On the other hand, in an in vitro study involving non-human primate blastocysts, human iPSCs were successfully incorporated into the developmental program of the monkey embryo in its early stages. These results have not been achieved in more evolutionary distant species, such as mice and pigs, which could be explained by xenogeneic barriers between human pluripotent stem cells and evolutionary distant host animal species [158]. There are other factors that make the progress of interspecies chimerism technologies difficult and that must be further studied, which may include species-specific differences in epiblast and trophectoderm development, developmental kinetics, and the maternal microenvironment, among others [161].

## 3. Regenerative Medicine in Ophthalmology

### 3.1. Overview

More than 2 billion people worldwide suffer from vision impairment [162]. Although ocular pathologies cover a wide range of distinct physiopathological pathways and etiologies, they can be analyzed by categorizing them anatomically. To this end, the eye can be divided broadly into two segments by the lens: the anterior segment and the posterior (vitreous) segment. Major structures of the anterior segment include the cornea, anterior chamber, iris, and lens. Meanwhile, the posterior segment is occupied by the vitreous body, retina, choroid, and sclera [163].

Drug delivery to these ocular structures represents a current challenge in ophthalmology. If possible, topical application would be the ideal route of application for both anterior and posterior segment diseases; nevertheless, the eye presents diverse barriers that do not allow for such conveniences. For instance, tear composition, turnover, and drainage, act as dynamic barriers to overcome, while corneal epithelium tight junctions and conjunctival absorption act as major static barriers for drug delivery. Moreover, the posterior segment poses an equal challenge to surmount. In the retina, for example, the blood–retinal barrier limits the entrance of systemically administered drugs into ocular tissues [164].

In a similar manner to drug therapeutics, RM therapies need to overcome several of the challenges that the eye imposes for their delivery. RM therapies have generally been applied directly into or adjacent to the tissues that they are intended to regenerate. For anterior segment conditions, topical, stromal, subconjunctival, transconjunctival, and perilimbal routes have been used. In the case of lacrimal gland disease, direct injection has been performed. Additionally, subretinal, intravitreal, suprachoroidal, subtenon, retrobulbar, and peribulbar transplantation have been carried out for posterior segment diseases.

Most RM in ophthalmology studies have focused on the use of stem cells for the treatment of limbal stem cell deficiency (LSCD), age-related macular degeneration (AMD), retinitis pigmentosa, corneal ulcers, and Stargardt’s disease, among other ocular pathologies [165]. By way of illustration, Holoclar^®^, the first authorized stem cell-based ocular therapy, has been recently approved in the European Union for the treatment of LSCD, with the use of autologous limbal cell transplantation [166].

A search in clinicaltrials.gov revealed a total of at least 209 interventional trials of RM in ophthalmology. Multiple early-phase I, phase I, I/II, II, II/III, and III studies of cell, exosome, scaffold, and gene-based therapies for eye diseases were found. These regenerative-based therapies are showing promise for the treatment of previously intractable and/or degenerative eye diseases, ranging from posterior segment pathologies such as AMD and retinitis pigmentosa to anterior structure diseases including Dry Eye Disease (DED) and LSCD. These therapies allow for approaches which are tailored specifically to each pathology’s etiology.

As seen in Table 1, most of these clinical trials for the treatment of ocular disease with regenerative therapies have been small, designed to demonstrate feasibility and safety, but have started to make headway towards an efficacy approach as many of these trials progress. This array of trials is in various states: (75) completed, (35) active not recruiting, (50) recruiting, and (49) in an assortment of statuses including suspended, withdrawn, unknown, not yet recruiting, enrolling by invitation, and terminated. The utilized therapies have been delivered by multiple routes, mainly subretinal and intravitreal injections, with an observable trend towards genetic therapy.

### 3.2. Published Evidence of Regenerative-Based Therapies in Ophthalmology

An analysis of the currently published clinical trials revealed subretinal drug delivery as a frequent route of administration for ocular diseases, particularly for posterior segment pathology, as well as the most prevalent mode of delivery overall. Although the subretinal route seems to provide potential safety benefits (such as reduced inflammatory reactions), its efficacy amongst the several regenerative-based therapies and diseases differs. Moreover, AMD was shown to be the most frequently published disease regarding regenerative-based therapy in ophthalmology.

Wet AMD, also known as exudative or neovascular AMD, is one of the major causes of central vision loss in the older age group in the urbanized and industrialized world. In the macular region (the area in charge of sharp and clear central vision), the delicate enzyme balance of the extracellular matrix is broken by aging retinal pigment epithelium cells (RPE). Senescent RPE cells trigger production of the vascular endothelial growth factor (VEGF), a proangiogenic factor, which, in turn, drives the development of choroidal neovascularization, where new vessels grow under or through the RPE via breaks in the Bruch membrane, the pentalaminar structure that usually separates the RPE and choriocapillaris [167]. This is the logic behind the therapeutic use of VEGF inhibitors such as ranibizumab, bevacizumab, or aflibercept, which imitate the antiangiogenic effects of naturally occurring substances such as angiostatin, endostatin, or soluble fms-like tyrosine kinase-1 (sFlt-1).

Reported effects of subretinal and intravitreal gene therapy observed among published clinical trials for Wet AMD included a decreased need for ranibizumab injections, improvement or maintenance of Best Corrected Visual Acuity scores (BCVA; the best possible vision that an eye can achieve with the use of glasses or contact lenses), and increase in aqueous humor levels of angiostatin and endostatin (antiangiogenic factors), among others [168,169,170,171,172]. The mechanism through which these effects were achieved range from the use of recombinant adeno-associated viruses (AAV) containing an anti-vascular endothelial growth factor agent (sFLt-1) to the use of lentiviral Equine Infectious Anemia Virus (EIAV) vectors expressing endostatin and angiostatin [168,172].

Regarding a different pathology, the previously mentioned adeno-associated virus vector, now containing the loaded RPE65 gene, has also proven useful for the treatment of Leber’s Congenital Amaurosis (LCA) type 2 when applied in the subretinal space. LCA is the result of a null mutation or biallelic loss of function in the RPE65 gene. Currently, this is the first and only gene therapy strategy approved by the Food and Drug Administration (FDA) for retinal degeneration to date (voretigene neparvovec-rzyl: Luxturna, Spark Therapeutics, PA, USA) [173]. The success of Luxturna in human clinical trials has paved the way for investigational studies targeting other genetic mutations associated with a variety of inherited retinal diseases.

Another highly studied retinal degenerative disease is Retinitis Pigmentosa, a heterogenous disease which is characterized by progressive degeneration of photoreceptors due to dysregulated apoptosis, with abnormal autophagic and necrotic signaling. Although initially only affecting rod photoreceptors (the sensitive cell responsible for scotopic and bichromatic vision), the destruction of large numbers of these structures has a deleterious effect on the RPE and begins to secondarily affect cone photoreceptors as well (the sensitive cell responsible for photopic and chromatic vision) [174,175,176]. Stem cell-based therapies could provide promising opportunities to repair the damaged retina and restore vision. For instance, FSCs, particularly retinal progenitor cells, can be collected and induced to differentiate into mature photoreceptors, which are then posteriorly subretinally transplanted into the affected eyes. Published clinical trials have reported results such as visual acuity improvement and maintenance and improved National Eye Institute Visual Function Questionnaire-25 scores (NEI VFQ-25, which measures vision-related functioning and the influence of vision problems on health-related quality of life). Other results included an improved fundus perimetry deviation index and electroretinography (ERG) parameters, temporary improvement in visual acuity and increase in retinal sensitivity of pupillary response, and tolerable safety profiles, among other outcomes, although results varied among disease genotypes [177,178,179,180]. On the other hand, in contrast to FSCs, applications of adult MSCs that have been investigated in preclinical and clinical studies, such as the use of intravitreal injection of BM-MSCs in patients with advanced RP, have thus far demonstrated results that have not been convincing [181].

Additionally, coculture transplantation of both fetal RPE cells and MSCs could possibly provide greater benefits compared to single-cell transplantation in retinal degeneration therapeutics. In a recent preclinical study, wild-type mice (C57BL/6J) were subjected to tail vein injections of 35 mg/kg sterile 1% sodium iodate (NaIO3) in saline in order to establish a retinal degenerative disease mouse model. Individually cultivated FSCs and MSCs that were then co-transplanted subretinally were found to relieve the atrophy of photoreceptors and preserve retinal function, evidenced by significantly improved electroretinogram results, as well as an increase in the survival rate of transplanted cells by suppressing immunoreaction and promoting the excretion of neurotrophic factors (observed through an increase in the expression of Crx, involved in activating rhodopsin and regulating rhodopsin levels), as well as a decrease in caspase-3 expression compared to mice who were transplanted FSCs or MSCs exclusively [182].

Regenerative-based therapies for Stargardt disease, the most common cause of juvenile macular dystrophy, are a good case in point for treatments studied among ophthalmological RM clinical trials that, although promising in their safety profile, have provided limited potential regarding possible benefits. Most of these clinical trials analyzed the use of human ESCs, with an overall observation that there was evidence of graft safety and cell integration and no evidence of hyperproliferation, rejection, or tumorigenicity, albeit with scant changes in visual acuity and the presence of areas of subretinal hyperpigmentation [183,184,185,186]. Gene therapy did not fare any better. Subretinal injection of the EIAV-ABCA4 vector (which contained the ABCA4 gene) in adults with Stargardt’s disease due to mutations in this gene demonstrated a lack of improvements in visual function, exacerbations of retinal pigment epithelium atrophy, and even chronic ocular hypertension [187]. These findings contrast those found in certain groups of wet and dry AMD, where human ESCs provided limited improvement in visual acuity, as well as positive cell integration and formation [188,189,190,191,192].

Continuing with a different RM therapy, exosomes have been used for the treatment of Dry Eye Disease in different clinical trials. DED is a multifactorial disease of the ocular surface characterized by a loss of homeostasis of the tear film and accompanied by ocular symptoms (such as blurry vision, stinging, burning, itching, and sensitivity to light, among others), in which tear film instability and hyperosmolarity, ocular surface inflammation and desepithelization, and neurosensory abnormalities play etiologic roles [193]. It was shown that the topical ophthalmic administration of MSC exosomes containing miR-204 (which directly targets interleukin-6 receptor to suppress the activation of the proinflammatory IL-6/IL-6R/Stat3 pathway) improved DED related to refractory GVHD by providing substantial relief in symptomatology, improving corneal epithelialization and increasing tear secretion [194]. Although these results provide a possibility for extended application of MSC exosomes in other severe dry eye diseases such as Sjogren’s syndrome and other connective tissue diseases, it still faces challenges for clinical applications, including its stability during storage and transport and the heterogeneity of exosome composition because of culture conditions and purification methods [195]. Perhaps these diseases could be approached from another regenerative therapy strategy. For example, a different study observed that a single injection of allogeneic ASCs generally improved both signs and symptoms in patients with severe aqueous deficient dry eye disease due to either primary or secondary Sjögren’s syndrome [196].

Potentially, exosomes could also be utilized to treat posterior segment pathologies such as glaucoma, the leading cause of irreversible blindness throughout the world and the second cause of reversible blindness after cataracts. Glaucoma can be described as a progressive optic neuropathy characterized by excavation or cupping of the optic disc, as well as apoptotic degeneration of retinal ganglion cells (RGC). The optic disc is the site where RGC axons coalesce and pass through the sclera and lamina cribrosa (a highly organized, multilayered, fenestrated connective tissue populated with astrocytes) to exit the globe as the optic nerve and relay visual information to the brain. The cup is the depression in the center of the optic disc, and in glaucoma, its progressive enlargement occurs due to damage in the lamina cribrosa and loss of retinal ganglion cell axons [197,198]. Pathological increase in intraocular pressure is the main risk factor for glaucoma onset. Intraocular aqueous drainage loss (due to stenosis of the anterior chamber, trabecular sclerosis, abnormal substances produced by extracellular matrix, or trabecular meshwork obstruction) can lead to an increase in pathological intraocular pressure, which, in turn causes optic nerve damage through mechanical compression and optic nerve ischemia. The longer the duration of intraocular pressure increase, the more severe the visual impairment [199].

It has been observed that exosomes derived from BM-MSCs may protect trabecular meshwork cells from oxidative stress; in animal models of glaucoma, MSCs promote the survival of RGC and their axons and preserve their function through miRNA-dependent mechanisms (especially through miRNA-17–92 and miRNA 21). For instance, the role of intravitreally administered UCMSC-derived exosomes (UCMSCs-exos) was explored in a rat model of optic nerve squeezing through exosome tracking, immunohistochemical analysis, fluorescence microscopy, etc., and concluded that they can promote the survival of RGCs but do not promote axonal regeneration. Staining by the GFAP antibody showed that the number of retinal glial cells treated by UCMSCs-exos increased and the activity was enhanced. On the other hand, another study found that non-pigmented ciliary epithelial cell exosomes (NPCE-Exos) induced a diminishment in the expression of two key canonical Wnt signaling proteins: pGSK3β and β-catenin. As an important component of the extracellular matrix, cadherin can increase the pore size of the trabecular meshwork, leading to an increase in the outflow resistance of the aqueous humor and an increase in intraocular pressure [104,199,200,201].

Lastly, limbal stem cell deficiency (LSCD) is one of the ocular surface diseases caused by a hereditary or acquired deficiency or loss of functional limbal epithelial stem cells in the corneoscleral limbus that supply the transparent corneal epithelium and, for this review, also serves as a successful case of a regenerative-based therapy application for anterior segment pathologies (at least among cases that have been published). This particular example is unique in so far as treatment of the disease’s profiles is already well established as being surgical, with the implantation and supplantation of corneal epithelial stem cells through autologous or allogeneic means, depending on various factors such as unilateral or bilateral affectation, wet or dry ocular surfaces, and severity, to name a few [202]. Other successful treatment modalities include the use of an autologous stem cell source of a different epithelial lineage, like oral or nasal mucosa, for patients with bilateral LSCD or the relatively novel approach of amnion-assisted implantation and cultivation [203]. As observed in Table 2, the results range positively from successful epithelial reconstruction and maintenance of re-epithelization to improvement of symptoms, quality of life, and visual acuity, among others [204,205,206,207,208]. Although treatment modalities that do not include a risk of rejection or the need for immunosuppression would be optimal, the current modalities offer many patients an opportunity at visual recovery.

### 3.3. Concluding Remarks

Although a detailed review of each published clinical trial extends beyond the scope of this analysis, it can be said that regenerative-based therapies have demonstrated promise in delivering curative therapies for previously intractable and/or degenerative eye diseases. There is an overall tendency towards gene therapy, and although the results seem generally positive, its associated immune and inflammatory reactions may render the treatment ineffective or harmful [243]. The severity of these reactions depends on the choice of vector and its route of administration, with subretinal delivery producing a weaker humoral response than the intravitreal route. This current review supports the concept that gene therapy is particularly useful for treating inherited diseases with loss of function mutations but that it can also be used to treat acquired diseases.

The reported clinical trials have also demonstrated the feasibility of administering stem cell therapies to the eye, both to the retina as well as to the vitreous. Although long-term data are not yet available and a relatively small number of patients have been treated, there are suggestions that implanted cells survive, are functional, and persist for months with encouraging measurable visual improvement for the patients [244]. There has been a continuing expansion in the number and types of stem cells assessed for potential use in ophthalmology, as seen by the use of embryonic stem cells, fetal stem cells, and adult stem cells amongst these clinical trials. Benefits of the therapy include the relatively small number of cells required, easy accessibility for surgery, and straightforward assessment and visualization of grafts. Limitations include immunological tolerance to transplanted cells (limitations reduced due to the use of autologous cells), tumorigenicity of transplanted stem cells, and ethical problems regarding the use and collection of certain cell types [245].

On the other hand, exosomes can be used as a therapeutic carrier to participate in multiple pathophysiological processes such as immune response, angiogenesis, and nerve repair in ocular-related diseases. Research on the role of exosomes in ocular-related diseases is still in its infancy, and although they possess novel benefits due to their unique miRNA cargo (with the possible opportunity to be modified and used as an organic nanovesicle), limitations include rapid clearance rates from the site of application, as well as a wide range of variability in exosome isolation which may influence study findings [199,246]. What can be said is that exosomes remain as an exciting frontier to explore potential therapeutics in different ophthalmological diseases.

Among other strategies that were touched upon in the text, such as gene editing and chimerism, they have somewhat limited roles in current ophthalmology and medicine. However, evidence of cellular function restoration and histological reconstruction demonstrate that these therapeutic strategies continue to show ongoing promise.

### 3.4. Future Directions

In conclusion, regenerative medicine is an evolving area that has gained renewed interest in recent years, with ophthalmology spearheading the progress among the field of medical specialties. The progress in the last decade alone is both vastly and measurably appreciable, with ever-progressing robust and scalable manufacturing processes, increasing adoption of expedited regulatory pathways, and new advances in therapeutic modalities and genomics, etc.

Although the current state of regenerative medicine research varies vastly in quality and quantity in regard to the research allotment that is allocated to distinct anatomic zones and/or vessels of application, it is clear that this effort will nevertheless sooner or later alter the patterns of conduct in ophthalmological treatment, thus expanding the field further [244]. Additional research is required to translate many of these encouraging experimental findings into clinical implementation and to develop standardized protocols for their utilization; as mentioned throughout the text, several challenges remain, including optimizing the differentiation protocols, ensuring transplanted cells’ safety and long-term viability, and developing effective delivery methods, among others [247].

## Figures and Tables

**Figure 1 cells-13-00179-f001:**
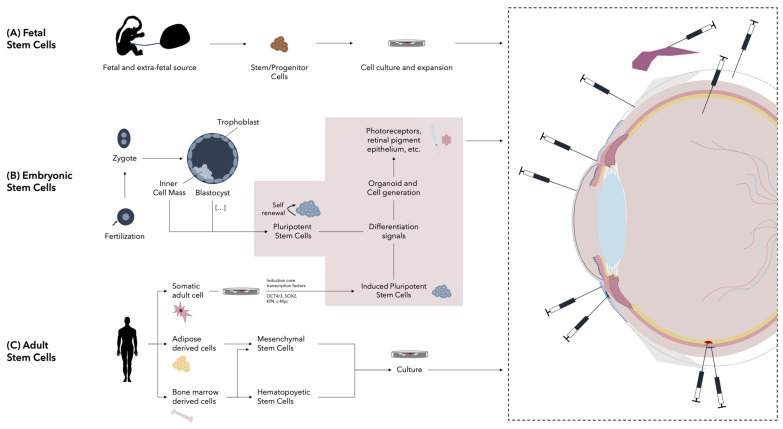
Stem cell classification based on origin and used administration routes of cellular therapies in ophthalmology. A broad representation of the most important stem cell sources is shown. (**A**) Fetal stem cells can be derived from fetal and extra-fetal sources. (**B**) Embryonic stem cells are pluripotent in nature and are obtained from the inner cell mass of the blastocyst. (**C**) Adult-derived stem cells can be obtained from several sources. The most important include bone marrow and adipose tissue. From bone marrow, hematopoietic stem cells and mesenchymal stem cells can be isolated. From adipose tissue, mesenchymal stem cells can also be found. Moreover, somatic adult cells can readily be induced into induced pluripotent stem cells, which as embryonic stem cells can give rise to all types of cells. As will be commented in the next section, cellular therapies have been applied in ophthalmology via several routes, including stromal, subconjunctival, intravitreal, subretinal, suprachoroidal, peribulbar, perilimbal, transconjunctival, subtenon, and lacrimal gland routes.

**Figure 2 cells-13-00179-f002:**
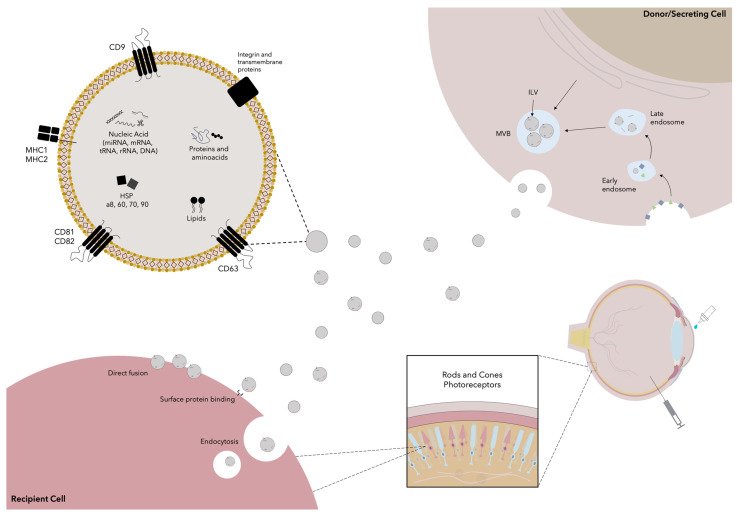
Exosome-based therapies. Exosomes are derived by secretion of intraluminal bodies (ILV) after the fusion of multivesicular bodies (MVB) with the plasmatic membrane. MVBs are derived from the maturation of early endosomes to late endosomes, which form several membrane invaginations that encapsulate the exosomes’ contents and give rise to the ILVs. Exosomes’ cargo includes several proteins, aminoacids, nucleic acids, and lipids that are received by the recipient cell after binding to surface proteins, which causes direct fusion of the exosome with the plasmatic membrane or its endocytosis. As will be mentioned in the following section, exosomes have been applied topically in clinical trials; nonetheless, there are high expectations for the use of exosomes intravitreally for the treatment of several retinal conditions.

**Figure 3 cells-13-00179-f003:**
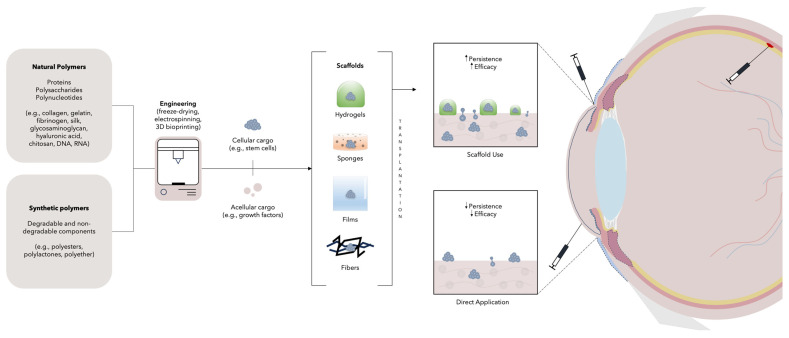
Scaffold-based therapies. Scaffolds can be synthetized from natural and synthetic polymers to form hydrogels, sponges, films, or fibers. Scaffolds can later hold both cellular (e.g., stem cells) and acellular (e.g., growth factors) cargo. After transplantation, scaffolds provide cells with an appropriate microenvironment for growth, improving the grafts’ persistence and efficacy. As will be commented in the next section, scaffolds have been used in different ophthalmology clinical trials and have been administered via stromal, perilimbal, and subretinal routes.

**Figure 4 cells-13-00179-f004:**
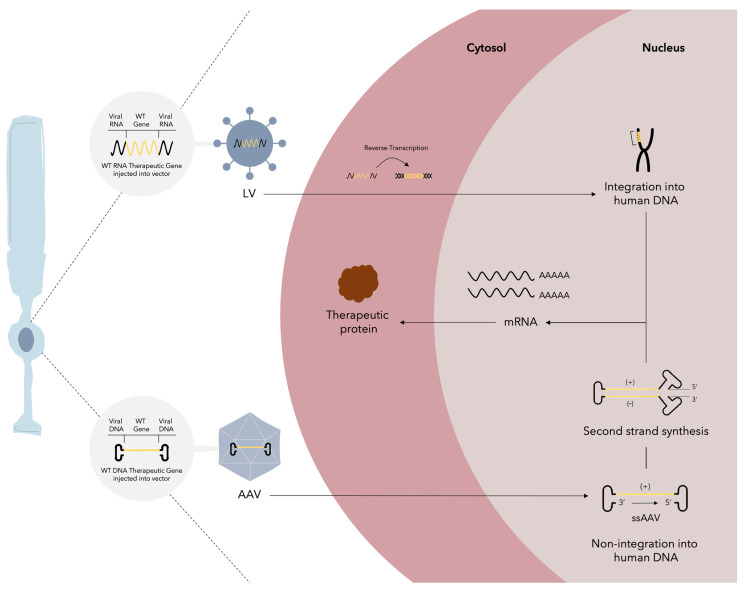
Gene therapy. The mechanisms by which this therapy functions are demonstrated in a photoreceptor. Integrating and non-integrating virus vectors can be used for gene delivery to the target cells. Lentiviruses (LV) are integrating RNA viruses that employ retrotranscription to generate their DNA, which is later integrated into the hosts’ genome for posterior mRNA transcription. Adeno-associated viruses (AAV) are single-stranded DNA viruses that do not integrate into the hosts’ genome, and after second strand synthesis, mRNA transcription is started. After transcription, mRNA is translated into the intended therapeutic protein. As commented in the next section, gene therapy has been used in ophthalmology for the treatment of different posterior segment diseases and has been applied via intravitreal and subretinal routes.

**Table 1 cells-13-00179-t001:** Registered clinical trials of regenerative medicine therapies in ophthalmology.

Anterior Segment					
Disease	Clinical Trials (n)	Strategy (n)	Administration Route (n)	Phase (n)	Status (n)
LSCD	26	LSCs (14) Oral mucosal epithelial sheets (6) Corneal epithelial cells (3) Scaffolds (2) BM-MSCs (1)	Perilimbal (26)	I (5) I/II (8) II (4) II/III (1) III (1) IV (1) NA (6)	Recruiting (5) Active not recruiting (2) Completed (12) Suspended (1) Withdrawn (1) Unknown (4) Not yet recruiting (1)
Corneal ulcer	5	BM-MSCs (2) ASCs (1) MSCs-secretome solution (1) Scaffold (1)	Subconjunctival (2) Topical (1) Perilimbal (1) Artificial cornea transplant (1)	I (1) I/II (1) II (3)	Recruiting (1) Completed (3) Not yet recruiting (1)
DED	6	ASCs (2) Exosomes (2) MSCs (1) Corneal epithelial cells (1)	Transconjunctival (1) Lacrimal gland (1) Topical (4)	Early I (2) I/II (2) II (1) NA (1)	Recruiting (1) Completed (2) Active not recruiting (1) Not yet recruiting (2)
KC	2	Scaffolds + ASCs (2)	Stromal (2)	Early I (1) II (1)	Recruiting (1) Unknown (1)
FECD	1	Gene therapy (1)	Intravitreal (1)	I (1)	Withdrawn (1)
BLAK	1	ASCs	Intralesional (1)	II (1)	Unknown (1)
Posterior segment					
Disease	n	Strategy (n)	Administration route (n)	Phase (n)	Status (n)
RP	56 *	BM-SCs (8) FSCs (13) hESCs (2) BM-HSCs (2) BM-MSCs (1) FSCs + Exosomes (1) HuRPE (1) Scaffolds (2) Gene therapy (25) Gene editing (1)	Subretinal (23) Intravitreal (21) Subtenon (4) Subtenon/IV/Retrobulbar (2) Subtenon/IV (1) Suprachoroidal (1) Peribulbar (2) Unknown (2)	Early I (3) I (11) I/II (22) II (8) II/III (5) III (5) NA (2)	Recruiting (12) Active not recruiting (13) Completed (18) Terminated (3) Unknown (7) Not yet recruiting (2) Enrolling by invitation (1)
AMD	52 *	hESCs (14) FSCs (7) BM-HSCs (1) BM-SCs (5) ASCs (1) iPSCs (2) Gene therapy (21) Gene editing (1)	Subretinal (33) Intravitreal (15) Suprachoroidal (1) Subtenon/IV/Retrobulbar (2) Subtenon/IV (1)	Early I (3) I (14) I/II (20) II (8) II/III (1) III (1) NA (5)	Recruiting (19) Enrolling by invitation (2) Active not recruiting (3) Completed (12) Withdrawn (3) Unknown (10) Not yet recruiting (3)
LCA	18 *	Gene therapy (17) HuRPE (1)	Subretinal (13) Intravitreal (4) Unknown (1)	Early I (1) I (3) I/II (10) II/III (2) III (1) NA (1)	Recruiting (2) Active not recruiting (5) Completed (7) Terminated (1) Enrolling by invitation (2) Unknown (1)
LHON	12 *	Gene therapy (11) BM-SCs (1)	Intravitreal (11) Subtenon/IV/Retrobulbar (1)	I (1) I/II (2) II/III (2) III (4) NA (3)	Recruiting (3) Active not recruiting (3) Completed (4) Not yet recruiting (1) Unknown (1)
CHM	10 *	Gene therapy (10)	Subretinal (9) Intravitreal (1)	I (1) I/II (3) II (4) III (2)	Active not recruiting (1) Completed (8) Enrolling by invitation (1)
SD	9 *	hESCs (5) BM-SCs (3) Gene therapy (1)	Subretinal (5) Intravitreal (2) Subtenon/IV/Retrobulbar (2)	I (2) I/II (4) II (1) NA (2)	Active not recruiting (1) Completed (3) Recruiting (1) Unknown (4)
ACHM	6	Gene therapy (6)	Subretinal (6)	I/II (6)	Recruiting (1) Active not recruiting (3) Completed (2)
DME	6 *	GT (6)	Intravitreal (5) Suprachoroidal (1)	Early I (1) I (2) II (2) III (1)	Completed (2) Recruiting (4)
DR	5 *	FSCs (1) BM-HSCs (1) GT (3)	Intravitreal (3) Suprachoroidal (1) Subtenon/IV (1)	I (2) II (3)	Recruiting (3) Active not recruiting (1) Completed (1)
BCD	2	Gene therapy (2)	Subretinal (2)	Early I (2)	Recruiting (2)
CRVO	2	BM-HSCs (2)	Intravitreal (2)	I (1) I/II (2)	Recruiting (1) Enrolling by invitation (1)
XR	2	Gene therapy (2)	Intravitreal (2)	I/II (2)	Active not recruiting (2)
Glaucoma	2	BM-MSCs (1) FSCs (1)	Intravitreal (1) Subtenon (1)	I (2)	Recruiting (1) Completed (1)
Macular holes	1	Exosomes (1)	Intravitreal (1)	I (1)	Active not recruiting (1)
TON	1	FSCs (1)	Unknown	III (1)	Completed (1)
AION	1	BM-MSCs (1)	Intravitreal (1)	II (1)	Active not recruiting (1)

Number of registered clinical trials (from clinicaltrials.gov) in ophthalmology by disease, therapy, administration route, phase, and status. * Clinical trials marked with asterisks include at least one trial that involved more than one ophthalmic disease. In these cases, the n is overestimated as one trial appears in multiple diseases. LSCD, Limbal Stem Cell Deficiency; DED, Dry Eye Disease; KC, Keratoconus; FECD, Fuchs Endothelial Corneal Dystrophy; BLAK, Bilateral Limbic-Associated Keratopathy; RP, Retinitis Pigmentosa; AMD, Age-related Macular Degeneration; LCA, Leber Congenital Amaurosis; LHON, Leber Hereditary Optic Neuropathy; CHM, Choroideremia; SD, Stargardt’s Disease; ACHM, Achromatopsia; DR, Diabetic Retinopathy; DME, Diabetic Macular Edema; BCD, Bietti’s Crystalline Dystrophy; CRVO, Central Retinal Vein Occlusion; XR, X-linked retinoschisis; TON, Toxic Optic Neuropathy; AION, Anterior Ischemic Optic Neuropathy; LSCs, Limbal stem cells; BM-MSCs, Bone marrow-derived mesenchymal stem cells; ASCs, Adipose tissue-derived Stem Cells; MSCs, Mesenchymal Stem Cells; hESCs, human Embryonic Stem Cells; FSCs, Fetal Stem Cells; BM-HSCs, Bone Marrow-derived Hematopoietic Stem Cells; BM-SCs, Bone Marrow-derived Mononuclear Stem Cells; iPSCs, induced pluripotent stem cells; HuRPE, human retinal pigment epithelium cells; IV, intravenous; NA, not applicable.

**Table 2 cells-13-00179-t002:** Published studies in regenerative medicine in ophthalmology.

Anterior Segment				
Therapy	Ocular Condition	Description	Main Outcomes	Ref.
Cell therapy	LSCD	Autologous and allogeneic stem cell therapy by corneal limbal epithelial transplantation in patients with LSCD.	Similar survival of autografts and allografts. High success rate, with a substantial improvement in symptoms, quality of life, vision, and epithelial quality.	[204]
		Allogeneic transplantation of corneal limbal stem cells cultured on an amniotic membrane in patients with bilateral LSCD.	Significant, sustained improvements in corneal epithelium, conjunctivalization, neovascularization, opacification, and conjunctival hyperemia were found. Significant improvements in visual acuity were also observed.	[205]
		Ex vivo, expanded autologous limbal stem cell transplant on human amniotic membrane in patients with unilateral LSCD.	Satisfactory ocular surface reconstruction occurred in all eyes. All patients showed an improvement in vision impairment and pain scores, while most experienced an improvement in visual acuity.	[206]
		Autologous or allogenic transplantation of limbal stem cells in patients with unilateral or bilateral LSCD, respectively.	Most patients were graded as anatomically successful based on the persistence of continuous epithelial surface. Only in patients anatomically successful was amelioration of visual acuity and pain found.	[209]
		Transplantation of a cultured autologous oral mucosal epithelial cell sheet in patients with LSCD.	Treatment was well tolerated, with 75% of patients having successful grafting after 360 days. Most patients showed an improvement in corneal ulcers and a decrease in the severity of punctate epithelial keratopathy.	[207]
		Transplantation of cultured autologous multilayered oral mucosal epithelium sheets in two patients with bilateral total LSCD.	Successful reversal of LSCD in the treated eyes was achieved for up to 24 months. Improvement in visual acuity and pain was observed.	[210]
		Transplantation of autologous cultured oral mucosal cell sheets in patients with total bilateral LSCD.	All treated eyes experienced a complete reepithelization of the corneal surface in the first week. A restoral of corneal transparency was shown, with a subsequent important improvement of visual acuity in all the treated eyes. During the 14 month follow-up, corneal transparency was maintained, and no complications were recorded.	[208]
		Transplantation of autologous cultured oral mucosal epithelial cell sheets in patients with total bilateral LSCD.	Most eyes achieved successful ocular surface reconstruction, with a complete stable epithelialization at 53.6 days on average. Some visual improvement was achieved. Expression of corneal cytokeratins in the grafts was demonstrated. No complications were observed.	[211]
		Trial comparing allogeneic BM-MSCs transplantation vs. cultivated limbal epithelial transplantation in patients with LSCD.	Both methods had similar high success rates (between 70% and 85%), with concurrent improvements in corneal epithelial phenotype. No adverse events related to the cell products occurred.	[212]
	KC	Transplantation of decellularized human corneal stromal laminas with or without ASC recellularization in the corneal stroma of patients with advanced keratoconus.	All patients had an improvement in visual parameters, refractive sphere, corneal thickness, and spherical aberration.	[213]
		Corneal stroma implantation of autologous ASCs in patients with advanced keratoconus.	No complications occurred. All patients had improved visual function. Central corneal thickness showed an improvement using corneal OCT. Patchy hyperreflective areas in the stroma demonstrated new collagen production. Survival of the implanted cells was confirmed with confocal biomicroscopy.	[214]
		3-year clinical outcomes of transplantation of decellularized human corneal stromal laminas with or without ASC recellularization in the corneal stroma of patients with advanced KC.	Significant improvement in visual acuity was observed in all groups but an increase in central corneal thickness was only observed in the groups which received a decellularized or recellularized stromal lamina but not in the one which received ASCs alone. No complications were observed at the 3 year follow-up.	[215]
		Implantation of ASCs, corneal decellularized laminas or ASCs recellularized corneal laminas in corneas with advanced keratoconus.	Overall gradual significant increase in anterior and posterior cellularity in patients’ stroma. The increase was significantly higher in the patients which received ASC recellularized laminas.	[216]
	DED	Injection of allogeneic ASCs into the lacrimal gland in patients with aqueous-deficient DED.	Improvements in DED symptoms, tear film stability and tear production occurred. No adverse events occurred.	[196]
		Intravenous infusion of allogeneic BM-MSCs in patients with refractory DED secondary to chronic GVHD.	An amelioration in symptoms was observed in more than half of the patients, along with increased tear secretion.	[217]
	BK	Anterior chamber injection of allogenous human corneal endothelial cells (CECs) supplemented with a ROCK inhibitor for the treatment of bullous keratopathy.	An increase in CEC density was found in all patients and most had an improvement in visual acuity of two or more lines.	[218]
Exosomes	DED	Topical administration of miR-204-containing MSC exosomes (as eye drops) in patients with refractory GVHD-associated DED.	Substantial relief in DED symptoms, reduced de-epithelization evidenced by fluorescein staining, improved tear quality and secretion evidenced by TBUT and Schirmers’ test, respectively.	[194]
Scaffolds	Corneal ulcer	Study protocol of bioengineered human allogeneic anterior corneas. The corneas were constructed with limbal epithelial cells and stromal fibroblasts from cadaveric donors using a biodegradable scaffold of agarose and fibrin. The constructed corneas were transplanted to patients with refractory severe trophic corneal ulcers.	NA. Only the study protocol was published, clinical results are pending.	[219]
Posterior segment				
Therapy	Ocular condition	Description	Main outcomes	Ref.
Cell therapy	Wet AMD	Subretinal (submacular) transplantation of hESC-derived human RPE cells in patients with wet AMD after removal of the neovascular membrane.	Follow-up at 12 months showed evidence of formation of a new RPE-like cell layer in damaged areas. Limited functional improvement was observed.	[188]
		Subretinal transplantation of an hESC-derived human RPE patch, consisting of an RPE monolayer on a coated synthetic basement membrane in patients with severe wet AMD.	Evidence of successful delivery of the RPE patch and a visual acuity gain of +20 letters over the 12 month follow-up.	[189]
	Dry AMD	Subretinal transplantation of hESCs-derived allogenic RPE cells (OpRegen) in patients with advanced dry AMD or geographical atrophy	Good toleration with no unexpected adverse events. Improvement in baseline visual acuity was found in some patients and persistence of the transplanted cells was suggested via imaging.	[190]
		1 year follow-up of the subretinal transplantation of an hESCs-derived RPE cell implant on an ultrathin parylene substrate in patients with advanced dry AMD.	More than half the patients reported at least one serious adverse event. No significant visual acuity improvements were observed, although some treated eyes experienced a >5-letter gain.	[191]
		Subretinal transplantation of an hESCs-derived RPE monolayer implant in patients with severe dry AMD.	Integration of the implant was demonstrated with OCT. Visual acuity did not improve in most patients, but no progression of vision loss was recorded in the implanted eyes.	[192]
		Suprachoroidal transplantation of autologous adipocytes, ASCs, and platelets in patients with dry AMD.	Patients were transplanted with all the three treatments. An improvement of more than 30% in visual acuity was observed at 180 days.	[220]
		SCOTS clinical trial for AMD. Different arms which combined retrobulbar, sub-tenon, intravitreal, subretinal, and intravenous administration of autologous BM-SCs in patients with dry AMD.	Most eyes showed a significant improvement in visual acuity (average 27.6%). No complications were observed.	[221]
		Subretinal transplantation of Human Central Nervous System Stem Cells (HuCNS-SCs) in patients with geographic atrophy due to dry AMD.	Changes in geographic atrophy areas were not found; nonetheless, the growth rate was significantly slower when compared with control eyes.	[222]
	SD and AMD	Subretinal transplantation of hESC-derived human RPE cells in patients with SD or AMD.	Good integration of cells into the host RPE layer. No signs of hyperproliferation, tumorigenicity, ectopic tissue formation or apparent rejection were found. Improvement in visual acuity was not clear, but little amelioration in visual acuity was found.	[223]
		22 month follow-up of subretinal transplantation of hESC-derived human RPE cells in patients with SD or AMD.	Evidence of medium-term to long-term graft safety and survival. No evidence of adverse proliferation, rejection, or serious ocular or systemic safety issues were found. Most patients had patches of increasing subretinal pigmentation. Visual acuity improved in some eyes.	[224]
		Subretinal transplantation of hESC-derived human RPE cells in patients with SD or AMD.	No evidence of adverse proliferation, tumorigenicity, ectopic tissue formation, or other safety issues was found. Visual acuity improved 9–19 letters in most patients.	[225]
		Subretinal transplantation of hESC-derived human RPE cells in patients with SD or dry AMD.	No adverse events related to the cell therapy were found. A significant improvement in visual acuity was found in AMD patients at 12 months. Improvements were also found in SD patients but due to the small sample size statistical analysis was not possible.	[226]
		Suprachoroidal implantation of ASCs in patients with dry AMD or Stargardt’s disease.	No systemic or ocular complications. Improvement in visual acuity, visual field, and multifocal ERG was found in all patients.	[227]
	SD	Subretinal transplantation of hESC-derived human RPE cells in patients with SD.	Dose-dependent development of areas of subretinal hyperpigmentation. No evidence of hyperproliferation or rejection. No significant improvements in visual acuity. Microperimetry found no evidence of benefit at the 12 month follow-up.	[183]
		Subretinal transplantation of hESC-derived RPE cells in patients with SD.	No serious adverse events were reported during the 3 year follow-up. Most patients did not show an improvement in visual acuity but treated eyes showed a slow-down in the progression of the disease.	[184]
		Subretinal transplantation of hESC-derived RPE cells in patients with SD.	No adverse events occurred within the 12-month follow-up; nonetheless, no significant increases in visual acuity were observed.	[185]
		5 year follow-up of hESC-derived RPE cells subretinal transplantation in patients with SD.	No long-term adverse events were noted. All operated eyes had a transiently increased or stable visual function 1–4 months after transplantation. Maintained morphological and functional changes were found in the RPE layer.	[186]
		SCOTS clinical trial. Different arms which combined retrobulbar, sub-tenon, intravitreal, subretinal, and intravenous administration of autologous BM-SCs in patients with SD.	Most eyes showed a significant improvement (average 17.96%) in average central vision. No adverse events were found.	[228]
	RP	Intravitreal injection of BM-MSCs in patients with advanced RP.	Several adverse events were found, such as posterior synechiae, cystoid macular edema, flat choroidal detachment, and intraocular lens displacement, none which remained at the 12 month follow-up. Slight improvement of visual acuity was found but returned to baseline within 12 months.	[181]
		Subretinal implantation of ASCs in patients with end-stage RP.	Most patients had ocular complications including choroidal neovascular membrane and epiretinal membrane. No significant improvements in visual acuity and ERG were observed.	[229]
		Intravenous infusion of UCMSCs in patients with advanced RP.	Visual acuity improved in most patients in the first 3 months and was improved or maintained for 12 months. The NEI VFQ-25 scores were significantly better during the first 3 months. No serious adverse effects occurred.	[177]
		Subretinal transplantation of fetal retinal progenitor cells in patients with advanced RP.	A significant improvement in visual acuity was observed in some patients; also, an increase in retinal sensitivity of pupillary response was shown between the 2 and 6 month follow-up. Nonetheless, these improvements faded at the 12 month follow-up. Integration of the transplanted cells was confirmed with OCT. No complications were reported.	[178]
		1 year follow-up of subtenon transplantation of UCMSCs in RP patients with different autosomal dominant or recessive and X-linked genotypes.	An improvement in outer retinal thickness was observed. Both autosomal dominant and recessive patients experienced a significant improvement in visual acuity, fundus perimetry deviation index, and ERG parameters at the 6 and 12 month follow-up, contrary to X-linked genotypes which did not. No complications were observed during the 1 year follow-up period.	[179]
	Retinal diseases	Intravitreal injection of autologous BM-HSCs in patients with irreversible ischemic or degenerative retinal conditions, including retinal vascular occlusion, hereditary or dry AMD, or RP.	No long-term ocular adverse events were noted. A slight improvement in visual acuity was recorded in most patients. Macular function in dry AMD patients worsened while the patient with retinal vascular occlusion showed a progressive improvement.	[230]
Gene therapy	Wet AMD	Subretinal injection of an rAAV.sFlt-1 adeno-associated viral vector containing an anti-vascular endothelial growth factor agent, sFLt-1, for the treatment of wet AMD.	rAAV.sFlt-1 was well tolerated, had a favorable safety profile, and decreased the need of ranibizumab injections.	[231]
		Combination therapy of subretinal ranibizumab and rAAV.sFlt-1 adeno-associated viral vector, which contains the sFLt-11 gene, for the treatment of wet AMD.	Patients receiving combination therapy required less ranibizumab injections than patients receiving ranibizumab alone (control group). BCVA was improved or maintained in 56% of patients in the combination group, compared to 36% in the control group. Adverse events were mainly procedure related and were self-resolved.	[168]
		Single subretinal injection of the rAAV.sFLT-1 adeno-associated viral vector, containing the sFLt-1 gene, for the treatment of wet AMD.	Adverse events were procedure related and were self-resolved. A slight visual acuity improvement was observed, and most patients did not require any anti-VEGF rescue injections in the 1 year follow-up.	[169]
		3 year follow-up of combination therapy of subretinal ranibizumab and rAAV.sFlt-1 adeno-associated viral vector for the treatment of wet AMD.	rAAV.sFLT-1 delivery was safe and well tolerated; nonetheless, no significant improvements or maintenance of visual acuity was found between the treatment and control groups.	[170]
		Single intravitreal injection of PF-04523655, a small interfering ribonucleic acid (siRNA) targeting the RTP801 gene, in patients with wet AMD.	PF-04523655 was generally safe and well tolerated. There were no dose-limiting toxicities. Efficacy of the treatment is not discussed.	[171]
		Subretinal injection of a lentiviral Equine Infectious Anemia Virus (EIAV) vector expressing angiostatin and endostatin (RetinoStat^®^) in patients with advanced wet AMD.	A dose-related increase in aqueous humor levels of angiostatin and endostatin was shown among patients, which was maintained in some patients at the 2.5 year follow-up, with some still showing expression at 4 years. The EIAV vector was shown to be safe although no significant changes in lesion sizes were found.	[172]
	SD	Subretinal injection of the EIAV-ABCA4 vector, containing the ABCA4 gene, in adults with SD due to ABCA4 mutations.	No improvements in visual function tests were noted. A subset of the treated eyes showed an exacerbation of retinal pigment epithelium atrophy. Chronic ocular hypertension, a serious adverse effect related to the treatment, occurred in one patient.	[187]
	RP	Single subretinal injection of the AAV8-coRPGR adeno-associated viral vector, containing the RPGR gene, in patients with X-linked R due to RPGR mutations.	A subset of patients demonstrated visual gains in the treated eyes, which were maintained up to six months. Retinal inflammation was observed in the patients that received the higher doses and was responsive to steroids. Other than the dose-dependent retinal inflammation, treatment proved to be safe.	[232]
		Subretinal injection of the rAAV2-VMD2-hMERTK adeno-associated viral vector, containing the MERTK gene, in patients with MERTK-associated RP.	rAAV2-VMD2-hMERTK injection was not associated with any serious side effects. A subset of patients reported improved vision on examination. One patient had a dramatic response to treatment, with a visual acuity of <20/6400, to 20/125 after a week, but worsened over time.	[233]
	LHON	REVERSE clinical trial. Single intravitreal injection of the rAAV2/2-ND4 adeno-associated vector, containing the mitochondrial ND4 gene, in patients with vision loss due to LHON.	Although the injection was only in one eye, sustained vision improvement was observed in both eyes. This finding suggested a transfer of the vector to the contralateral eye, which was later demonstrated in non-human primates.	[234]
		RESCUE clinical trial. Single intravitreal injection of rAAV2/2-ND4 adeno-associated vector, containing the mitochondrial ND4 gene, in patients with vision loss ≤6 months from onset due to LHON.	Both treated and untreated eyes’ visual acuity continued to deteriorate comparably. No significant improvements in visual acuity occurred.	[235]
		RESTORE study. Long-term follow-up study of RESCUE and REVERSE clinical trials. In both trials, a single intravitreal injection of rAAV2/2-ND4 was administered to LHON patients.	The analyses combined the results of both trials. A progressive and sustained visual acuity improvement from 12 to 51 months after vision loss onset was observed. A clinically meaningful improvement in quality of life was also shown.	[236]
	CHM	THOR trial. Subretinal injection of the AAV2-REP1 adeno-associated viral vector, containing the CHM gene, in patients with choroideremia.	Maintenance or minor improvements in visual acuity were recorded. Mean retinal sensitivity, peak retinal sensitivity, and gaze fixation area also improved in most patients. Adverse events were related to the surgical procedure.	[237]
		Subretinal (subfoveal) injection of AAV2-hCHM adeno-associated viral vector, containing the CHM gene, in patients with choroideremia.	No vector-related or systemic toxicities were noted. No improvements in visual acuity, visual sensitivity, nor in rate of disease progression were observed. Serious adverse events occurred and included acute foveal thinning and macular hole.	[238]
		Subretinal (subfoveal) injection of AAV.REP1 adeno-associated viral vector encoding the REP1 gene in patients with choroideremia.	An improvement in visual acuity was observed in most patients, with the more severe cases being the ones which benefited the most. An improvement in mean retinal sensitivity was also noted and was dose dependent.	[239]
	DME	Intravitreal injection of PF-04523655, a small interfering ribonucleic acid (siRNA), targeting the RTP801 gene, compared to laser photocoagulation, in patients with diabetic macular edema.	The injection of PF-04523655 was generally safe and well tolerated and, in comparison to laser photocoagulation, showed a dose-related greater improvement in visual acuity. No serious adverse events were related to the siRNA treatment.	[240]
	LCA	Subretinal injection of the AAV2.hRPE65v2 adeno-associated viral vector, carrying the RPE65 gene, in patients with LCA.	There was a significant improvement in the pupillary light reflex in the treated eyes. There was also an improvement in visual acuity, visual field area, and a decrease in nystagmus. There were no serious adverse events.	[241]
	ACHM	Subretinal injection of the AAV8.CNGA3 adeno-associated viral vector, containing the CNGA3 gene, in patients with CNGA3-associated achromatopsia.	Minor but significant improvements in visual acuity, contrast sensitivity, and color vision were recorded. Treatment demonstrated a good safety profile.	[242]

Comprehensive list of published clinical trials utilizing regenerative medicine strategies for the treatment of ophthalmologic conditions. DED, Dry Eye Disease; BC, Bullous Keratopathy; AMD, Age-related Macular Degeneration; SD, Stargardt’s Disease; RP, Retinitis Pigmentosa, LSCD, Limbal Stem Cell Deficiency; KC; Keratoconus; DME, Diabetic Macular Edema; LHON, Leber Hereditary Optic Neuropathy; LCA, Leber Congenital Amaurosis; CHM, Choroideremia; ACHM, Achromatopsia; MSC-Exos, Mesenchymal Stem Cell-derived exosomes; MSCs, Mesenchymal Stem Cells; BM-MSCs, Bone Marrow-derived Mesenchymal Stem Cells; BM-SCs, Bone Marrow-derived Mononuclear Stem Cells; ASCs, Adipose tissue-derived Stem Cells; hESCs, human Embryonic Stem Cells; HuCNS-SCs, Human Central Nervous System Stem Cells; BM-HSCs, Bone Marrow-derived Hematopoietic Stem Cells; FSCs, Fetal Stem Cells; UCMSCs, Umbilical Cord Mesenchymal Stem Cells; GVHD, Graft Versus Host Disease; RPE, Retinal Pigment Epithelium; NEI VFQ 25, National Eye Institute Visual Functioning Questionnaire-25; OCT, Optical Coherence Tomography; ERG, Electroretinography; TBUT, tear break-up time.

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
