# Peer review of "Beyond Vision: An Overview of Regenerative Medicine and Its Current Applications in Ophthalmological Care"

_cells, 2024, doi:10.3390/cells13020179_

Round 1

Reviewer 1 Report (Previous Reviewer 1)

Comments and Suggestions for Authors

In my previous round of review I asked the following items for improving the article that none of them are considered in the new submission and unfortunately other reviewers commnts from the 1st round are not considered too : 

Here is the list of my comments:

The article up to page 16 discusses general topics that have been published in similar texts in the past and I recommend that it be deleted.

From page 16 onwards, the article can be classified based on the subtitles before page 16 to determine what treatment methods can be performed in ophthalmology based on the treatment titles (which are to be deleted in the section that is to be deleted). From page 16 onwards, it needs a structure.

Table 2 can be divided into several tables and included in new substructures.

General images 1 to 4 can also be dedicated for ophthalmology.

The article is not suitable for publication in its current form and with a lot of additional content.

Author Response

  1. The article up to page 16 discusses general topics that have been published in similar texts in the past and I recommend that it be deleted.

R: Thank you for your comment. As replied to your previous valuable comment, the objective of our review is to both describe regenerative medicine strategies and then explain their uses in the ophthalmological field. Moreover, there are therapies that have not yet been used in ophthalmology but may have future applications in the field, which are also explained in this section (e.g., epigenetic reprogramming, organoids, chimeras). With this we intend to inform readers about the different therapies in regenerative medicine before indulging them into the ophthalmology applications to avoid confusion. Due to this, we consider the therapies section of our manuscript to be essential, we are sorry to know that our objective went unnoticed. Nonetheless, with this in mind, the therapies section was summarized even further, especially in the cell therapy section.

  1. From page 16 onwards, the article can be classified based on the subtitles before page 16 to determine what treatment methods can be performed in ophthalmology based on the treatment titles (which are to be deleted in the section that is to be deleted). From page 16 onwards, it needs a structure.

R: Thank you for your suggestion. As previously replied, the ophthalmology section in the manuscript is generally arranged by disease, a method which we consider to be of advantage than rather sectioning by therapy as most of the ophthalmological disorders we have mentioned have been treated with more than one regenerative medicine therapy. Moreover, sectioning by disease allows for the description of each condition. Nonetheless, in response to your valuable suggestion, we have diligently restructured Table 2 by therapy, striving to align with your feedback.

  1. Table 2 can be divided into several tables and included in new substructures.

R: Thank you for your suggestion. As mentioned previously, we have rearranged Table 2 in a manner that it is sectioned by therapy more than by disease.

  1. General images 1 to 4 can also be dedicated for ophthalmology.

R: As previously replied, figures 1 to 4 have the objective of generally representing the therapies, not yet specifically in ophthalmology, to support the text explaining the therapies. Nonetheless, figures 1 to 3 (as figure 4 was already representing gene therapy in a photoreceptor) were adapted to ophthalmology and the figure footers have been updated accordingly.

Reviewer 2 Report (Previous Reviewer 4)

Comments and Suggestions for Authors

This resubmitted review covers the major approaches currently used in regenerative medicine targeted towards the eye. As the abstract states, this article is a very high-level review of stem cells, EVs, and other current methods, which makes the review somewhat difficult for a reader to appreciate given its large size and expansive nature with limited mechanistic details. As mentioned by another reviewer, the article would benefit from limiting the very general sections about stem cells that have been thoroughly reviewed in the literature. The authors were also not responsive to the suggestion regarding excluding the oversimplified figure of the eye and common injection sites from the article.

Comments on the Quality of English Language

Minor editing needed throughout.

Author Response

  1. As mentioned by another reviewer, the article would benefit from limiting the very general sections about stem cells that have been thoroughly reviewed in the literature.

R: Thank you for your suggestion. We have further limited the cell therapies section by removing or summarizing sections that were extensive but not essential to present the different therapies.

  1. The authors were also not responsive to the suggestion regarding excluding the oversimplified figure of the eye and common injection sites from the article.

R: Thank you for your comment. We considered this figure important in order to represent the eye anatomy and administration routes to non-ophthalmologist readers. Nonetheless, we agree that it is oversimplified and does not present essential information. Figure 5 has now been deleted from the text.

Reviewer 3 Report (Previous Reviewer 2)

Comments and Suggestions for Authors

None

Author Response

Thank you for your insight.

Round 2

Reviewer 1 Report (Previous Reviewer 1)

Comments and Suggestions for Authors

I do not have more suggestion.

Reviewer 2 Report (Previous Reviewer 4)

Comments and Suggestions for Authors

No additional comments.

Comments on the Quality of English Language

Minor modifications needed.

This manuscript is a resubmission of an earlier submission. The following is a list of the peer review reports and author responses from that submission.

Round 1

Reviewer 1 Report

Comments and Suggestions for Authors

The article up to page 16 discusses general topics that have been published in similar texts in the past and I recommend that it be deleted.

From page 16 onwards, the article can be classified based on the subtitles before page 16 to determine what treatment methods can be performed in ophthalmology based on the treatment titles (which are to be deleted in the section that is to be deleted). From page 16 onwards, it needs a structure.

Table 2 can be divided into several tables and included in new substructures.

General images 1 to 4 can also be dedicated for ophthalmology.

The article is not suitable for publication in its current form and with a lot of additional content.

Reviewer 2 Report

Comments and Suggestions for Authors

“Beyond Vision: An Overview of Regenerative Medicine and Its Current Applications in Ophthalmological Care” is well-written, informative, and contains clear and well-structured figures. However, there are a few areas that require attention and revision:

Introduction: The introduction section is somewhat lacking in content related to ophthalmology, focusing more on the introduction of regenerative medicine strategies. This results in a noticeable disconnect between the introductory section and the subsequent "Regenerative Medicine in Ophthalmology" section. To enhance the coherence of the article, I suggest expanding the introduction to include a brief overview of regenerative medicine's relevance and significance in the field of ophthalmology.

Section Ordering: There seems to be a typographical error in the ordering of sections. In line 365, "Very Small Embryonic-Like Stem Cells" (2.1.3.2.5) appears after "Induced tissue-specific stem cells" (2.1.3.4). This should be corrected for logical flow and consistency.

Gene Therapy Section: The discussion of gene therapy in lines 762-767 appears to be somewhat loosely connected to its relevance to Wet AMD. While it is mentioned in relation to retinal degeneration, it might be beneficial to provide a more direct link between gene therapy and its application in addressing Wet AMD. This could include discussing specific gene therapy approaches for Wet AMD or highlighting the potential impact on this specific condition more explicitly.

Abbreviation: In line 726, the abbreviation for "Bone Marrow-derived Hematopoietic Stem Cells" is incorrectly presented as "BM-HSCs." It should be revised to "BM-HSCs" to ensure consistency and accuracy in the use of abbreviations throughout the manuscript.

Reviewer 3 Report

Comments and Suggestions for Authors

Overview: This is a well written, resourceful review article covering an important field of medical science, now and more in the future. I have only a few concerns as mentioned in the specific comments below.

Specific comments

Abstract: It needs reorganising different points in the order of background/purpose, methods, results and conclusions although these subheadings are not required to be written.

Figure 2 is not linked to the text. Please do it.

Figure 5 (L680): Is there any difference between subconjunctival and transconjunctival injection, or even perilimbal?

It would be good if the authors added future directions for the regenerative medicine.

Reviewer 4 Report

Comments and Suggestions for Authors

This review article provides a summary of different regenerative methods used in ophthalmology with a focus on cell-based approaches. Overall, the manuscript is expansive and well-written and the figures are very descriptive and informative. The review would benefit by limiting the scope and focusing more on specific areas of regenerative medicine (stem cells or EVs, etc), rather than covering every area. Suggestions to improve the quality of the manuscript are included below.

-(line 415 - 416): Exosomes are described as '...microvesicles...', which is not accurate. Please correct throughout the manuscript. 

-(pg. 6 - lines 252-254): The authors mention that the low recovery of BM-MSCs from tissues is a disadvantage. Further subculturing will lead to a higher cell count given that these are stem cells thus overcoming much of this issue. Further elaboration should be included in the text.

-Please modify the superscript for cell numbers throughout the manuscript (ex: lines 267-268).

-Fig. 5 includes a schematic of the eye and common injection sites. This figure is over-simplified and likely not needed in this review article.

Comments on the Quality of English Language

Minor editing needed throughout. Re-formatting of superscripts are needed throughout the text.